# Sentinel Lymph Node Detection in Breast Cancer: An Innovative Technique

**DOI:** 10.3390/diagnostics13122030

**Published:** 2023-06-12

**Authors:** Paolo Izzo, Claudia De Intinis, Simone Sibio, Luigi Basso, Andrea Polistena, Raimondo Gabriele, Massimo Codacci-Pisanelli, Luciano Izzo, Sara Izzo

**Affiliations:** 1Department of Surgery “Pietro Valdoni”, Policlinico “Umberto I”, Rome “Sapienza” University of Rome, 00128 Rome, Italy; deintinis.1891513@studenti.uniroma1.it (C.D.I.); simone.sibio@uniroma1.it (S.S.); luigi.basso@uniroma1.it (L.B.); andrea.polistena@uniroma1.it (A.P.); raimondo.gabriele@uniroma1.it (R.G.); massimo.codacci@uniroma1.it (M.C.-P.); luciano.izzo@uniroma1.it (L.I.); 2Unit of Colorectal Surgery, Department of Medical, Surgical, Neurologic, Metabolic and Ageing Sciences, University of Campania “Luigi Vanvitelli”, 81100 Caserta, Italy; sa_izzo@hotmail.it

**Keywords:** SLNB, SPIO, ALND, LS, superparamagnetic iron oxide nanoparticles

## Abstract

(1) Background: Sentinel lymph node biopsy is important in the search for metastases, especially in patients with malignant breast disease. Our study proposed new techniques to prevent complications such as possible postoperative seroma formation, pain or hypoesthesia of the axillary cord and medial arm surface, as well as motor deficits, to avoid disabling outcomes and presents initial data from our experience with the sentinel lymph node biopsy technique. (2) Methods: We mainly used two radioactive tracer detection techniques and a new technique using a radiotracer called Sentimag-magtrace. The positive lymph node was located and removed to perform histologic analysis. In our study, we evaluate 100 patients who underwent breast cancer surgery. (3) Results: We calculated the identification rates of the different methods of sentinel lymph node detection and found that it was 88.9% using radioactive tracers vs. 89.5% using the magnetic tracer technology (Sentimag). (4) Conclusions: Thus, this technique avoids radiation exposure for both patients and health care providers, and can reduce costs and time.

## 1. Introduction

Biopsies of the sentinel lymph node (SLN) have a high importance for metastases research, especially in patients with malignant breast disease, where the metastatic spreading generally occurs through the lymphatic system in a progressive manner or without jumps of lymph nodes; this technique has high reliability, and allows the patient to avoid an axillary lymph node dissection (ALND) and all of its complications [1,2].

The various lymphocytes can be found in aggregates in lymphoid tissues, which give the tissues their name. When these aggregates are found in the mucous membranes’ connective tissue, which is generally abundant in the cavities of the body that communicate with the external environment, such as the respiratory, digestive, urinary or reproductive tracts, the lymphoid tissue is called mucosa-associated lymphatic tissue (MALT). In some areas of the body, lymphocytes and macrophages can join to form aggregates called lymphatic or follicle nodules, from which the lymphocytes recirculate, ready to activate an immune response. These nodules are characteristic of lymph nodes, tonsils and the appendix.

The most interesting lymphoid organs for the SNL technique are obviously the lymph nodes; these are the largest lymphoid organs, scattered throughout the body, that carry out the filtration functions of the lymph and are the activation sites of B and T lymphocytes. The lymph node is a bean-shaped oval organ that usually does not exceeds three centimeters in length and on one of its sides, it converges to form a recess. The lymph nodes are covered by a connective fibrous capsule to which the sectors called trabeculae are attached which deepen to divide the organ into compartments. Between the capsule and the parenchyma, there is a small space called the subcapsular breast which contains reticular fibers, or collagen fibers, macrophages and dendritic cells. The parenchyma is divisible into a cortical part, which covers almost the entire circumference of the lymph node, and a more internal medullary part.

The cortical part is mainly composed of lymphatic nodules that, in a state of activation against a pathogen, are separated from germinative centers where the B lymphocytes multiply and differentiate into plasma cells. The medullary parts is composed of a sort of network of mid-collar cords composed of lymphocytes, plasma cells, macrophages, reticular cells and reticular fibers. Furthermore, both the cortical and the medullary parts have respective breasts in continuity with the subcapsular breast.

The lymph enters the lymph nodes on their convex surface through the different ventilated manifold vessels, flows into the subcapsular breast and then slowly through the breasts of the cortex and medulla where, by contacting the of the present immune cells, it is filtered by the various extraneous substances; later, the lymph comes out through the only effective lymphatic vessel, which, together with blood vessels, emerges from the hilium.

The lymph nodes, due to their role as stations along the course of a vessel, can be reached by metastases as well as pathogenic agents. The metastatic diffusion of tumors is the phenomenon in which neoplastic cells detach from the primary tumor, and spread to other blood and/or lymphatic organs. They continue to proliferate uncontrollably, determining the formation of secondary neoplasms. Due to the high permeability of lymphatic capillaries, neoplastic cells easily penetrate and spread through the lymph. They then localize in the regional lymph nodes where they proliferate, replacing the lymphoid cells; once the neoplasia is well implanted in a lymph node, the cells can further spread to other lymph nodes.

However, if the metastasis is detected early enough, the malignant pathology can, in some cases, be eradicated by removing not only the primary tumor, but also the nearest loco-regional lymph nodes. For example, a malignant breast pathology is often treated with the removal of the neoplastic mass through a partial or total mastectomy, combined with the removal of the axillary lymph nodes also called axillary lymphadenectomy (ALND). However, this technique has many complications; the main one is represented by lymphedema of the upper limb (abnormal accumulation of lymph, which can be temporary or permanent) caused by the interruption of lymphatic vessels that drain the flow of lymph from the limb itself, with an incidence of about 10%.

Other complications include the possible formation of a post-operative seroma, pain or hypoesthesia of the axillary cavity and medial surface of the arm, as well as motor deficits; the latter may be due either to accidental nerve injury during surgery or, more frequently, to the formation of fibrous scar tissue that limits limb movement. It is precisely in order to prevent the debilitating outcomes of ALND that the technique of sentinel lymph node biopsy becomes relevant.

The sentinel lymph node is the first lymph node that receives lymph from a tumor and is therefore likely to be the first to be reached by metastases. Through biopsy and histological analysis of this lymph node, it is possible to determine its involvement in metastasis and consequently refer the patient to axillary lymph node emptying. Metastases detected by histological examination can be classified into macrometastases, if they have a diameter greater than or equal to 2 mm, micrometastases if they are between 0.2 and 2 mm, and isolated tumor cells (ITCs) in case of a total diameter of less than 0.2 mm. The aim of our study was to investigate the use of new techniques to prevent complications such as the possible formation of a post-operative seroma, pain or hypoesthesia of the axillary cord and medial surface of the arm, as well as motor deficits to avoid disabling outcomes; here, we present the first data of with the use of the sentinel lymph node biopsy technique [2,3].

## 2. Materials and Methods

The study was approved by the Sapienza University Research Federated Athensis (No. C26F09B5E5). In our study, we considered 100 patients who were diagnosed with breast cancer after mammography and histological examination. The inclusion criteria were (1) any patient who needed to localize an intractable breast lesion; (2) any patient who required a sentinel lymph node biopsy; (3) the patients were collected from January 2019 and January 2021. Comparisons were made between patients with different localization methods: with radiative tracer or with magnetic tracer technology (SentiMag). Baseline demographic data were collected at the time of surgery and the outcomes were recorded. Staging, lymph node status, lymph node detection rates and complications were collected (Figure 1).

### 2.1. Vital Dye Technique

Vital dyes are substances that are injected into the lymph nodes and allow their localization by presenting a bluish coloration. There are various types such as isosulfan blue, methylene blue and sulphan blue, although the most widely used is Patent Blue V which consists of an internal salt or calcium compound of diethylammonium hydroxide.

The dye can be injected subdermally or peritumorally. In the first case, after the induction of general anesthesia, 0.2–0.4 mL of Patent Blue V is injected into the skin projection of the neoplasm, and a gentle massage of the injected area is performed to facilitate the progression of the dye towards the axillary cavity. In the second case, it is necessary to inject larger volumes (2–4 mL) in two or more peri-tumoral points and it is also necessary to massage the injection area to obtain a more rapid diffusion of the dye towards the axillary cavity. However, the subdermal route is preferred because the smaller volume injected reduces the skin tattoo that is created by diffusion from the injected area, making it easier to find the sentinel lymph node; moreover, the rich lymphatic vascularization of the dermis allows the dye to reach the sentinel lymph node more quickly.

The removal of the lymph node can begin after about 15 min from the injection, guided by finding the areas that have been dyed blue.

This method is less used but still very effective in detecting the sentinel lymph node; important contraindications are the blue marks that it leaves for a long time on the patient’s skin and also hypersensitivity reactions of the anaphylactic type which has a probability of about 2% [3,4,5] (Table 1).

Inclusion Criteria:Patients needing to localize an intractable breast lesion;Patients requiring a sentinel lymph node biopsy;Patients recruited from January 2019 to January 2021.

Data Collected:-Baseline demographic data at the time of surgery;-Staging information;-Lymph node status;-Lymph node detection rates;-Complications.

### 2.2. Radioactive Tracer Technique

In this technique, the radiopharmaceutical 99mTecnezio Nanocoll is used; this consists of colloidal particles of human albumin, with a size between 20 and 80 nm (nanocolloids), labeled with the metastable radioisotope technetium 99 (99mTc). Sentinel lymph node identification can be performed at two time points: before the operation by means of a breast lymphoscintigraphy and during the operation by means of a radio-guided surgery probe. Lymphoscintigraphy can also be performed the day before the operation or the same day at least 1–2 h before the operation.

The radioactive tracer (99mTc) is injected by the nuclear physician, generally with a dose of 0.2 milliCuries (mCi) (in patients with higher body weights, the dose is increased to 0.4 mCi), in a solution ranging from 0.2 to 0.4 mL in volume, using a 25-gauge needle. An excessive volume of the injected radiopharmaceutical can collapse the lymphatic capillaries and hinder accumulation in the sentinel lymph node.

In the case of palpable nodules, the localization is simple, while in case of non-palpable lesions, the injection is based on neoplastic opacities detected by ultrasound guidance or on an aggregate of tumor microcalcifications detectable under mammographic stereotactic guidance.

Basically, the injection can be performed in three modalities:-Subdermal injection: this injection occurs in the subcutis at the lesion and allows for a faster migration of the radiopharmaceutical, taking advantage of the rich lymphatic vasculature of the dermis.-Perilesional injection: intraparenchymal injection around the mass, usually performed in deep neoplasms.-Periareolar injection.

Intratumoral inoculation is not recommended because inflammation is often created around the lesion and the 99mTc nanocolloid tracer is unable to pass through this layer of inflamed tissue.

After the injection, a gentle massage is given to the site to facilitate lymphatic drainage and the radiocolloid is given time to reach the lymph node.

At this point, lymphoscintigraphy can be performed using a gamma camera, a device found in nuclear medicine departments, which converts the gamma photons emitted by the decays of the radionuclide inside the patient into images that can be viewed on a monitor.

Immediately after the administration of the radiopharmaceutical, it may be useful, but not mandatory, to document the initial kinetics of tracer distribution using a dynamic acquisition of 10–15 frames over 1 min; after about 15 min, static planar acquisitions are made in two or three projections but in case of poor migration of the radiocolloid, the images are also acquired after 3 and 18 h.

The first and most revealing projection is the 45° anterior oblique (right or left depending on the breast affected by the lesion, or both if both are involved); it is acquired by keeping the surface of the gamma camera head as parallel as possible to the axillary cavity and to better distinguish the inoculation point from the sentinel lymph node, which will appear as the first node draining the neoplastic tissue. The second is the anterior projection, which is important for the lymph nodes of the internal mammary chain. Finally, lateral projection images can be acquired.

After the acquisitions, with the help of a Cobalt57 point source (57Co), the skin projection of the same lymph node is highlighted and marked with an indelible ink.

The lymphoscintigraphic image available before surgery is of great value as it allows the detection of the location and the presence of one or more lymph nodes to direct the surgeon before surgery.

The radiopharmaceutical used for sentinel lymph node biopsy (SLNB) is 99mTc Nanocoll, which consists of nanocolloidal particles of human albumin labeled with metastable technetium-99. Nanocoll is prepared using human serum albumin obtained from human blood donors, tested in accordance with European Community regulations and found to be non-reactive for hepatitis B surface antigens (HBsAgs), human immunodeficiency virus antibodies (anti-HIV 1/2) and hepatitis C virus antibodies (anti-HCV). These particles, with a maximum diameter of 80 nm, easily enter the lymphatic capillaries, and are thus drained and captured by the lymph nodes. In rare cases, hypersensitivity reactions may occur.

99mTc is a metastable isotope of technetium (metallic element with Z = 43) produced by the decay of molybdenum-99 (99Mo). From a physical point of view, 99mTc has optimal characteristics as it is a pure gamma emitter with a photon energy of 140 KeV (ideal for imaging with a gamma camera/SPECT), has a half-life of 6 h (long enough to be transported from a production site to a target site and short enough to reduce the dose to the patient and their companions) and is easily available. Chemically, 99mTc is eluted from the generator column as technetium pertechnetate (99mTcO4-).

The production of 99mTc is based on the use of 99Mo/99mTc generators. The generator is a system by which a parent radionuclide with a relatively long half-life (about 67 h for 99Mo) produces a child radionuclide with a short half-life (6 h for 99mTc). They consist of a cylindrical lead container that contains a column of ion exchange resin; the different adsorptions on the column of parent and child radionuclides allows the isolation, after appropriate elution, of the child radionuclide. The elution of the column is performed using a solvent, in this case a physiological solution, which allows the separation of the child radionuclide from the column without altering the bond of the parent radionuclide to the column itself, thus obtaining an eluate with a high radiochemical purity. The column is connected to the outside by two small tubes which, starting from the two ends of the column, end up in the needles fixed in two cavities at the top of the generator. To elute the 99mTc, a glass vial with a pierceable rubber cap is inserted into one of the two needles, which contains sterile physiological solution; another vial is inserted into the second needle, which is placed in a shielded container (lead or tungsten), in which an air vacuum has been created. The depression created by the vacuum causes the emptying of the vial containing the physiological solution that, after interacting with the resin column, removes the 99mTc that, at the end of the elution, will be contained in the second vial. Immediately after elution, the resin column contains only 99Mo; since the 99Mo → 99mTc decay continues, new 99mTc starts to form immediately (Table 1).

### 2.3. Magnetic Tracer Technology

In recent years, another method has been developed, which does not involve the use of radioactive material for the identification of the sentinel lymph node (LS), which is currently used in all identifications; instead, a non-radioactive magnetic tracer is used representing a real revolution in the approach to this procedure. This system consists of the Sentimag probe and the Magseed magnetic marker. The marker is placed under ultrasound guidance up to 30 days before surgery. During surgery, we use Sentimag to precisely localize the Magseed and thus the lesion. This new tracer (Figure 2), which is commercially available only under the name of Sienna+ ^®^, consists of a black-brown aqueous suspension composed of nanoparticles of superparamagnetic iron oxide (SPIO superparamagnetic iron oxide) coated with carboxydextran that must be used only in conjunction with a Sentimag^®^ magnetometer [6,7]; the coating of carboxydextran, while maintaining biocompatibility, prevents the agglomeration of these particles that have a diameter of about 60 nanometers, which is small enough to be drained into the lymphatic capillaries.

The procedure is similar to that with the radioactive tracer but only in the operative phase. A volume of about 5 mL, consisting of 2 mL of magnetic tracer and 3 mL of saline solution, is injected in the periareolar or perilesional area and for 5 min the inoculation area is massaged in order to speed up the lymphatic drainage. After about 20 min (in the case of elderly or overweight patients, the migration of particles requires more time), the search for the sentinel lymph node is initiated by passing the magnetometer probe along the axillary cord. Once a discrete signal is detected, an incision is made and the search for signal is continued from the inside; access can also be made by exploiting a previous surgical breach.

The SentiMag instrument exploits the principle of magnetic susceptibility, generating an alternating magnetic field that transiently magnetizes the iron oxide particles of Sienna+; these respond with an induced magnetic field of low intensity, with the same direction as the applied external field, (superparamagnetism) which is then detected by the probe itself.

The presence of the nanoparticles will then be detected by producing a sound and a signal on the display of the device, whose intensity will be proportional to the intensity of the induced magnetic field that increases with proximity (Figure 3) to the iron particles, which have collected mainly in the sentinel lymph node.

The surgeon can locate the LS based on the signal strength (Figure 4) as well as the visually brown coloration that the magnetic tracer gives to the lymph node; it can then be removed and a histological examination can be performed. This process is repeated until any remaining lymph node has a signal reading of less than 10% of the first LS removed.

This magnetic tracer cannot be used in patients with hemosiderosis (iron overload disease) or with previous hypersensitivity reactions to dextran compounds [8].

In a multicenter study of SentiMag that was the first to evaluate the magnetic technique for sentinel lymph node biopsy (SLNB) (Figure 5), it was compared to the standard procedure (radiopharmaceutical with or without vital dye). It was demonstrated to have a capacity of LS identification equivalent to the method using radioactive tracer (94.5% Sienna+ vs. 95% 99mTc). In the MagSNOLL study, it was shown that, after intratumoral injection, the magnetic tracer allows the localization of non-palpable breast lesions. Another important property of Sienna+ is that it possesses characteristics similar to the superparamagnetic contrast agent used in MRI; this means that once it is injected, lymph nodes could be visualized through an MRI of the axillary cavity, in which an inhomogeneous uptake of Sienna+ in the sentinel lymph node can help identify a metastatic formation, thus replacing preoperative breast lymphoscintigraphy [9,10] (Table 1).

## 3. Results

Our study consisted of 100 patients (median age 62 ± 6 years, ranging from 55 years old to 70 years old) who underwent surgery for T1a, b, and c or T2 stage (less than 2.5 cm) breast cancer. Table 2 shows the patient population characteristics. The readings obtained from the two methods show that the data obtained are comparable (Table 3). We performed sentinel lymph node diagnostics using traditional scintigraphy for 60 and Sentimag for 40 identifications (Table 3). The main complication was lymphedema with an incidence of about 10% (Table 4).

## 4. Discussion

The sentinel lymph node (LS) technique is performed in those patients with biopsy-proven malignant breast disease that meet certain criteria, without which, the examination would not be useful. Most of them can be classified by the TNM classification, which is an international system of classification of malignant tumors, from which the stage of the disease can be derived [11,12].

This classification consists of three main parameters as indicated by the name:-T parameter indicates the size of the primary lesion and can take values from 1 to 4. It can be measured in situ and indicates if an intraepithelial carcinoma that has the morphological characteristics of malignancy but without the infiltrating capacity.-The parameter N indicates the status of the lymph nodes close to the tumor, and is 0 when they are completely unharmed, otherwise it can be 1, 2 or 3 with increasing severity.-Parameter M indicates the presence of distant metastases and can be 0 (no metastases) or 1 (presence of metastases).-A further parameter G can have a value from 1 to 4, indicating the degree of differentiation of malignant cells, that is, to what extent they have maintained the characteristics of the cells from which they originated.

An X value may also be assigned to the first three parameters to indicate the inability to define that characteristic. For sentinel lymph node biopsies (SLNB), the general recommendation is to reserve the method to women with monocentric T1–T2 breast neoplasms less than 3 cm in diameter and with clinically negative axillary lymph nodes [13,14]. This aspect can be detected by an ultrasound of the axillary cavity, which is able to highlight the presence of lymphadenopathy; however, this method is very operator-dependent because its outcome is related to the experience of those who perform the examination. In the case of structural alterations or doubtful results, a cell sample is taken and subjected to histological examination to identify its nature. This sampling is called ultrasound-guided cytology if it is performed by ultrasound-guided needle aspiration (FNAC, fine needle aspiration cytology), which involves the sampling of a few cells through a fine needle; if one or more pieces of tissue are taken through a thicker needle, it is called biopsy and can be performed under ultrasound guidance (CB, core biopsy) or under stereotactic mammographic guidance (mammotome).

Excellent results over the years have made it possible to perform SLNBs in cases of multifocal neoplasms but only if the overall diameter of the foci is less than 3 cm and always with clinically negative axillary lymph nodes (the multifocality of a neoplasm is the presence of multiple foci in the same breast quadrant).

Patients undergoing conservative surgery with sentinel node biopsy should undergo standard follow-up, i.e., with a visit initially every six months, and with annual mammography. The periodic examination should include an accurate clinical examination of axillary lymphoglandular stations; if one or more lymph nodes are found to be enlarged, an axillary ultrasound and a possible needle aspiration may be useful. In the case of recurrence of disease in the axilla (3–4% of cases), it is necessary to resort to delayed complete axillary lymphadenectomy [14,15].

In cases where the sentinel lymph node is positive on histological examination, axillary dissection is performed; in order to reduce motor limitations of the upper limb, cycles of physiotherapy can be performed.

In cases where the sentinel lymph node is negative, adjuvant radiation therapy is used to reduce the probability of local recurrence of the tumor.

As mentioned several times, the sentinel node technique can reduce the incidence of upper limb lymphedema compared to complete axillary dissection. However, it should be noted that the incidence of lymphedema after delayed lymphadenectomy, due to false-negative sentinel nodes or to recurrence of disease in the axilla, is about 17% higher than after primary lymphadenectomy, probably because an area that has already been surgically manipulated is re-operated [15].

## 5. Conclusions

Axillary lymph node status still remains the most important prognostic factor in malignant breast neoplasia. Selective sentinel lymph node (LS) biopsy was developed to assess axillary lymph node status [16] and avoid unnecessary surgery (axillary lymphadenectomy) in patients with histologically negative lymph nodes for metastases. At present, ALND is performed in all cases of positive LS, although some studies raise the debate whether it should be performed in patients with micrometastases and isolated tumor cells.

The most widely used technique for LS individualization is the one using a radioactive tracer (99mTc), which has a high accuracy and a very low probability of false negative results. Compared to an ALND, this technique involves fewer complications for the patient (lymphedema, motor deficits, etc.), lower costs and execution times, preservation of immunocompetent tissue and a psycho-physical improvement of the patient. The main disadvantage is the compulsory presence of a nuclear medicine service to perform the lymphoscintigraphy and produce the radiopharmaceutical; another disadvantage is the exposure to ionizing radiation, although minimal, that the use of radioisotopes entails [17].

The method with the magnetic tracer is a valid alternative to the use of radioactive tracers, especially for those hospitals that do not have a nuclear medicine section; it can produce an enormous reduction in costs and time by avoiding the scintigraphic procedure before and during the operation. It also avoids all the disposal procedures of materials contaminated by radioactive substance and the compulsory closure times of the operating room required by law. This method does not imply a loss of information since the scintigraphic examination can be replaced by one using the magnetic detector; in this way, there would be a complete elimination of the exposure to ionizing radiation both to the healthcare team and the patient [18]. This will also reduce the time required for the patient’s diagnostic procedure, thus decreasing their psychological stress, which is already of considerable importance, since they are faced with a malignant pathology [19] (Table 5).

## Figures and Tables

**Figure 1 diagnostics-13-02030-f001:**
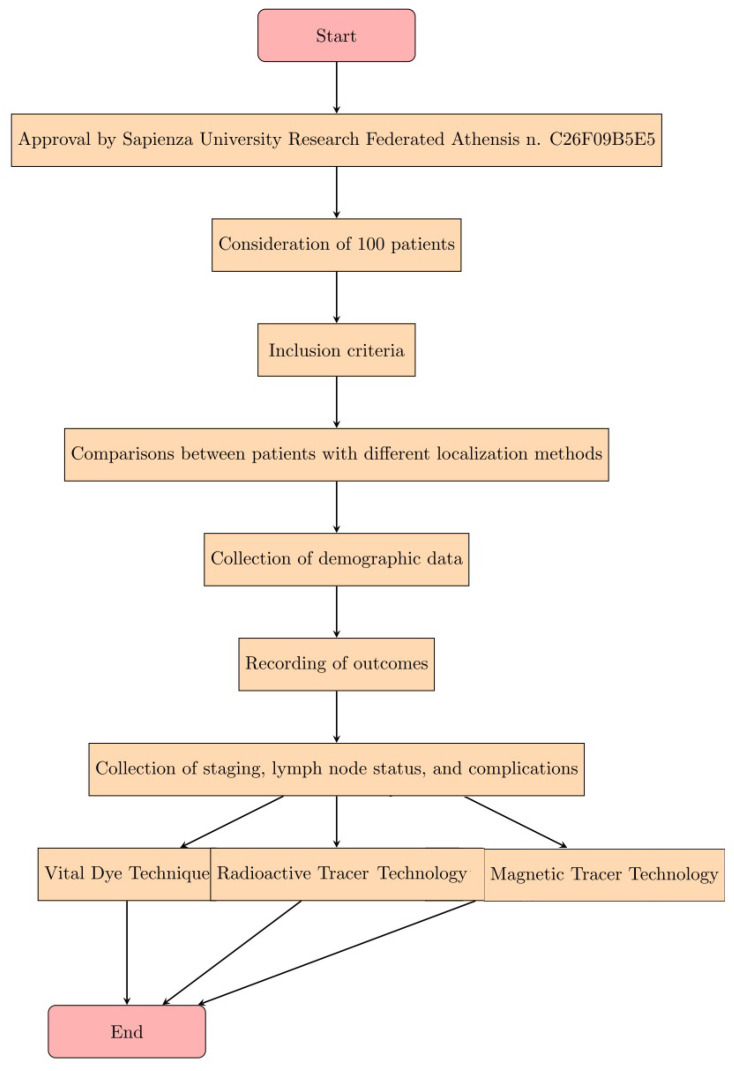
Expression of the steps of the Materials and Methods in the form of a diagram.

**Figure 2 diagnostics-13-02030-f002:**
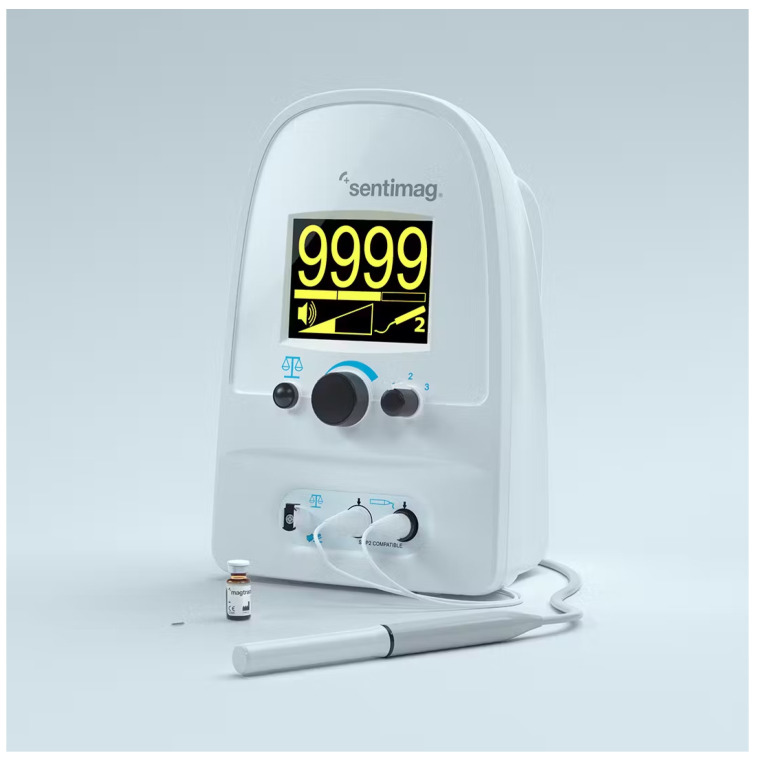
“Sentimag” handheld magnetometer device integrated with Sienna+ tracer.

**Figure 3 diagnostics-13-02030-f003:**
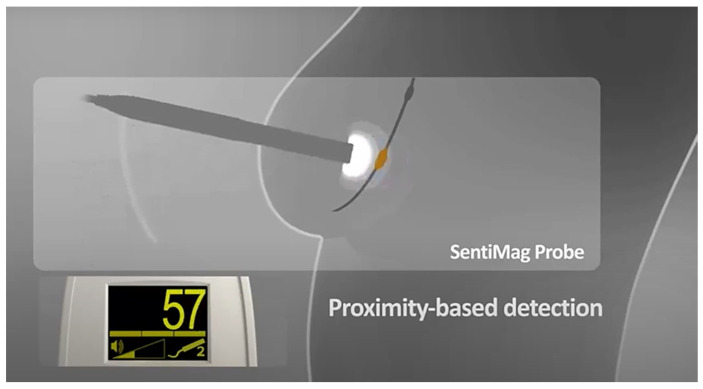
Principle of proximity-based detection of Sentimag device.

**Figure 4 diagnostics-13-02030-f004:**
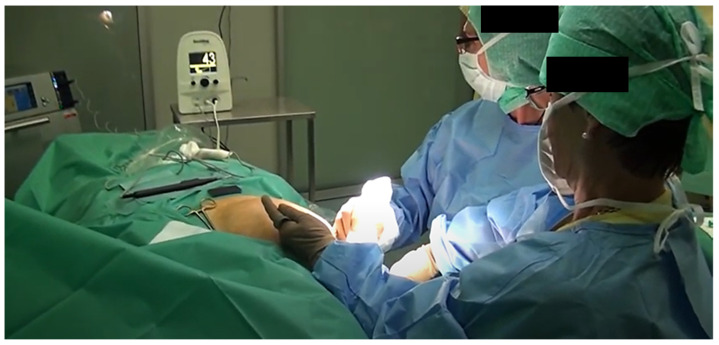
Sentimag application: localization of the LS in the area with the greatest signal.

**Figure 5 diagnostics-13-02030-f005:**
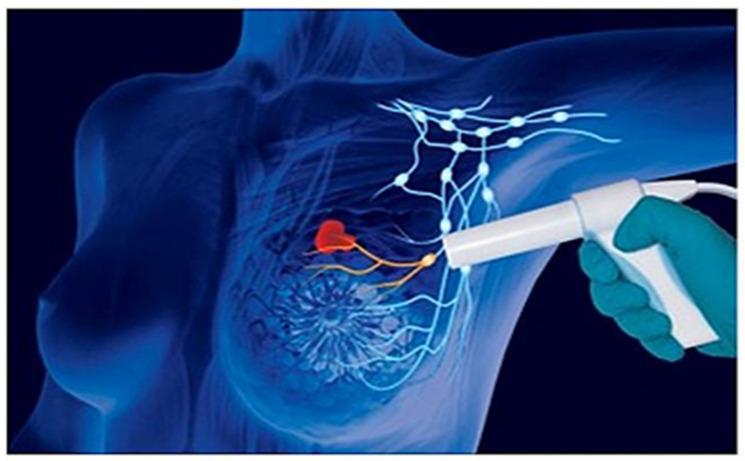
Experimental technique of SLNB through the use of the “Sentimag” magnetometer.

**Table 1 diagnostics-13-02030-t001:** Comparison of Localization Methods for Sentinel Lymph Node Biopsy in Breast Cancer Patients.

Localization Method	Technique Description	Advantages	Disadvantages
Vital Dye Technique	Injection of vital dyes into lymph nodes to provide blue coloration	Effective in detecting sentinel lymph node	Blue marks on skin, hypersensitivity reactions
Radioactive Tracer Technique	Injection of radiopharmaceutical (99mTc Nanocoll) into lymph nodes	Precise localization with lymphoscintigraphy, dynamic and static planar acquisitions	Radioactive material, requires specialized equipment
Magnetic Tracer Technology	Injection of magnetic tracer (Sienna+) into lymph nodes	Non-radioactive, precise localization with Sentimag magnetometer, visualization with MRI	Cannot be used in patients with hemosiderosis, hypersensitivity to dextran compounds

**Table 2 diagnostics-13-02030-t002:** Patient population characteristics.

Age at Cancer Diagnosis (years; mean)	45.3 ± 10
Menopausal status (%)	
Menopause	44.8
No menopause	55.2%
Age at menarche (years, mean)	13.6 ± 3.8
Lactation for at least 3 months (yes, %)	43%
Mammographic breast density (%)	48.9%
BI-RADS 1–2 (no dense breast)	
BI-RADS 3–4 a, b, and c (dense breast)	51.1%
Suspicious microcalcifications	16.4%
Discordance (>10 mm) between mammographic and US detection of the main index lesion and/or its dimensions (%)	12%
Positive for BRCA1/2 (%)	2.4%
First-degree family history of breast carcinoma (%)	23.2%
Discordance (>10 mm) between mammo-graphic and MR detection of the main index lesion and/or its dimensions (%)	0.7%

**Table 3 diagnostics-13-02030-t003:** Identification rates of methods.

Technique	Identification Rate %	Number of Patients
Radiative tracer	88.9	60
Magnetic tracer technology (Sentimag)	89.5	40

**Table 4 diagnostics-13-02030-t004:** Complications.

Technique	Complication	Percentage
Radiative tracer	upper limb lymphedema	15%
Magnetic tracer technology (Sentimag)	upper limb lymphedema	4%

**Table 5 diagnostics-13-02030-t005:** Comparison of Techniques for Sentinel Lymph Node Biopsy in Malignant Breast Neoplasia.

	Techniques	
	Radioactive Tracer (99mTc)	Magnetic Tracer
Accuracy	High	High
Cost	Higher	Lower
Execution Time	Longer	Shorter
Nuclear Medicine	Required	Not required
Ionizing Radiation	Present, minimal exposure	Eliminated
Disposal	Required	Not required

## Data Availability

The data presented in this study are available on request from the corresponding author.

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
