# Peer review of "Sentinel Lymph Node Detection in Breast Cancer: An Innovative Technique"

_diagnostics, 2023, doi:10.3390/diagnostics13122030_

Round 1
Reviewer 1 Report
The authors have mentioned a diagnostic method for breast cancer. The overall article is good, but it needs corrections before publication
1. Express the steps of the mentioned material method in the form of a diagram.
2. The numbering of the tables should be corrected. Only one table is numbered.
3. In line 307, Table1: refer to other diagnostic methods and about advantages and disadvantages.
4. In the last table, move the technique column to the appropriate place and you can use the item instead. Modification of the table is required.
Author Response
Dear reviewer,
as you suggested:
1) I expressed the steps of the "Methods and Matherial" section in the form of a diagram;
2) I numbered all the table and I added the corresponding reference in the article;
3) as for your third suggestion, the table number 1 that I created referred to the techniques mentioned in the article, to which the table refers. Therefore, I would like to keep the table with the current informations;
4) I modified the last table as you required.
To facilitate the review of changes, I used the "Track Changes" function.
I hope now everything it's ok.
Thank you very much. Best regards.

Reviewer 2 Report
Article very interesting, I would like to ask which are the improvements this d techniques gives? and which is the cost and is possible to apply in all the hospitals?. Is better this technique that the centinel node PCR detection?
